# Clinical Efficacy of Hyaluronic Acid with Iodine in Hard-to-Heal Wounds

**DOI:** 10.3390/pharmaceutics15092268

**Published:** 2023-09-01

**Authors:** Jana Pecová, Vladimíra Rohlíková, Markéta Šmoldasová, Jan Marek

**Affiliations:** 1Medical Faculty, Masaryk University in Brno, 62500 Brno, Czech Republic; 2Vsetín Hospital, 75501 Vsetín, Czech Republic; 3Hospital in Ústí nad Orlicí, 56218 Ústí nad Orlicí, Czech Republic; 4Long-Term Care Facility Albertinum Žamberk, 56401 Žamberk, Czech Republic

**Keywords:** chronic wounds, ulcer, treatment, hyaluronic acid, iodine

## Abstract

Hard-to-heal wounds do not heal spontaneously and need long-term care provided by specialists. That burdens the patients as well as the healthcare systems. Such wounds arise from several pathologies, which result in venous leg ulcers (VLU), diabetic foot ulcers (DFU), pressure ulcers (PU), or ulcers originating from post-surgical wounds (pSW). Given the complex nature of hard-to-heal wounds, novel treatments are sought to enable wound healing. We tested the clinical efficacy and applicability of fluid comprising hyaluronic acid and iodine complex (HA-I) in the treatment of hard-to-heal wounds. Patients (*n* = 56) with VLU, DFU, PU, or pSW hospitalised in multiple wound-care centres in the Czech Republic were treated with HA-I. Wound size, classically visible signs of infection, exudation, pain, and wound bed appearance were monitored for 12 weeks. The highest healing rate was in DFU (71.4%), followed by pSW (62.5%), VLU (55.6%), and PU (44.4%). Classical visible signs of infection were resolved within 8 weeks in all types of wounds. Wound bed appearance improved most noticeably in pSW and then in VLU. Exudation was lowered most significantly in DFU and pSW. The highest decrease in pain was in pSW and DFU. The treatment with HA-I successfully led to either complete closure or significant improvement in the wound’s healing. Therefore, the complex of hyaluronic acid and iodine is suitable for the treatment of hard-to-heal wounds of various aetiologies.

## 1. Introduction

Skin wound healing is a complex process aimed at the recovery and functionality of this type of soft tissue. It occurs in several overlapping phases, including inflammation, granulation (the formation of new tissue), vascularized, and scar maturation. However, several underlying pathologies, such as diabetes, venous insufficiency, post-surgical complications, and other comorbidities, may induce the formation of hard-to-heal (aka chronic) wounds [1]. Such wounds do not heal spontaneously within 8 weeks; their healing takes several weeks or longer, even when treated by a specialist.

Hard-to-heal wounds may contain areas of black necrotic tissue or be covered with fibrin slough. Inflammation is connected with high exudation and pain. When hard-to-heal wounds heal, exudation ceases, signs of inflammation regress, pain decreases, and the wound appearance changes to contain vascularised, vivid-red-coloured granulation tissue and activated epithelial wound margins and islets. There are several strategies for supporting the healing of hard-to-heal wounds. Many of them include the application of advanced wound dressings containing hyaluronic acid [2].

Hyaluronic acid (HA) is a natural polysaccharide that is an essential compound of the extracellular matrix (ECM), including that of skin and healing wounds [3]. HA plays an essential role during skin wound healing [4]. It is deposited early in the haemostasis stage of wound healing from platelets, adjusts the mechanical properties of the scab, and mediates the migration of cells into the wound. High molecular weight (HMW) HA present in ECM is cleaved during inflammation into smaller fragments with different signalling properties than HMW HA [5,6]. Such fragments, for example, potentiate the angiogenesis of newly formed tissue. Fibroblasts produce large quantities of HA early in wound healing, which contributes to the structural and functional properties of newly formed granulation tissue. HA also stimulates keratinocytes to migrate over the wound and to reestablish the epidermal barrier. Given the irreplaceable role of internal (produced by cells) HA in wound healing, this molecule is used to support the healing of hard-to-heal wounds in the form of advanced materials. HA can be processed into various forms, including sheets or fluids used in chronic wound healing [7]. Externally applied hyaluronic acid accelerates the healing of hard-to-heal wounds and decreases wound pain [8,9]. Reepithelization of HA-treated wounds is faster. Besides its active signalling properties, HA maintains a moist wound environment for favourable wound healing, helps to manage excess wound exudate, reduces dressing adherence to the wound, and can act as a carrier of active compounds such as antiseptics. 

One factor that hinders the proper healing of hard-to-heal wounds is bacterial infection. Wound bacteria shelter themselves from immune cells in the EPS matrix, form biofilms, produce virulence factors, and aggravate ongoing inflammation [10]. The efficacy of antibiotics is rapidly reduced for the biofilm bacteria, which may enable them to gain antibiotic resistance. Therefore, wounds are often treated with antiseptics that have broad-spectrum efficacy. 

Iodine has been used against microorganisms for about 200 years (Lugol’s solution was introduced in 1829; Davies mentions iodine use in wound treatment in 1839 [11]). Iodine is a potent antiseptic active against Gram+ and Gram- bacteria, viruses, fungi, or spores. Despite its long use, microbes did not develop resistance to iodine, unlike chlorhexidine, triclosan, or silver [12,13]. Although several iodine forms exist, only molecular iodine (I_2_) is considered the active antimicrobial iodine species [14]. However, I_2_ is poorly soluble in water, while its salt, KI, is readily soluble in water. Therefore, one of the first iodine formulations was a water solution of I_2_ and KI (I_2_ + I^−^ → I_3_^−^). Later, iodine was combined with iodophors (such as PVP, cadexomer, etc.), which improved the kinetics of iodine release [14,15]. Iodine release from the fluid may be slowed down due to its viscosity or ionic interactions of I_2_ with HA. The formal oxidation state of I_2_ in Hyiodine is lowered by its reaction with I^−^. In PVP-I, the formal oxidation state of I_2_ remains 0, since I_2_ interacts with a lone electron pair of oxygen. Therefore, Hyiodine may irritate the wound bed less than PVP-I.

The complex of hyaluronan, molecular iodine, and potassium triiodide (KI_3_) in the form of viscous fluid is regarded as an advanced dressing for chronic wounds [2]. In vitro results show that hyaluronan decreases the toxicity of the KI_3_ complex [16]. In addition, this complex decreased the PMA (phorbol 12-myristate 13-acetate)—induced oxidative burst of blood phagocytes. Ex vivo results in rat wounds underline the unique properties of hyaluronan complexed with iodine. The KI_3_ complex with hyaluronan accelerated the early healing of acute rat wounds and provided for a moist wound environment [17,18]. The fluid was previously reported to aid the healing of diabetic ulcers (14 of 18 patients healed within 6 weeks) [19], venous leg ulcers, decubiti, and post-operative complicated wounds such as abdominal dehiscences [20]. The HA-I fluid was used to successfully treat sternal dehiscence, a rare but severe cardio-surgery complication [21]. The KI_3_ complex either aided in preparing the wounds for plastic surgery or helped heal the wound. Recently, this antiseptic fluid was reported to help clean war wounds before delayed sutures [22]. 

This study investigated in detail treatment efficacy with HA-I (Hyiodine) fluid in hard-to-heal wounds of different origins. We report healing rate, pain, exudate management, and wound bed formation.

## 2. Materials and Methods

### 2.1. Materials

The HA-I fluid used throughout this study (Hyiodine) is a mixture of high molecular weight (>1 MDa) hyaluronic acid (1.5%), potassium iodide (0.15%), and iodine (0.1%) manufactured in Contipro, a.s. (Czech Republic).

### 2.2. Study Design

A post-market clinical follow-up study (PMCF) was conducted in multiple wound-care centres. A total of 10 clinical sites in the Czech Republic that specialise in chronic wound healing were selected from private practices, regional hospitals, and university hospitals across the country. This was an open-label study. Since this study focused on treatment efficacy and applicability to various types of complicated wounds, there were no control treatments. The intended study length was until the wounds healed or until 12 weeks of treatment. Treatment could be discontinued before the end of the study at the discretion of the physician in cases where the patient’s health significantly deteriorated and prevented the continuation of the study. Several patients were dismissed, and the wounds were treated at home; therefore, the records are missing.

### 2.3. Inclusion and Exclusion Criteria

The inclusion criteria for the patients were as follows: age > 18 years; complicated, possibly deep wounds not healing for more than 6 weeks, previously either treated or untreated with other wound device(s). Patients with known sensitivity to iodine, thyroid gland dysfunction, or kidney dysfunction were excluded from this study. Patients with malignant or autoimmune wounds and systemic immunosuppression were also excluded from this study.

### 2.4. Patient Enrolment and Characteristics

The demography of the patients and the aetiology of the wounds are summarised in Table 1. 

### 2.5. Treatment Regimen

The attending physicians and nurses were instructed on how to apply the fluid to the wounds. It can be administered directly to the wound bed via a sterile syringe. Indirect application, which is suitable for wounds more than 4 cm in diameter, is performed via a sterile dressing carrier (such as sterile gauze). The fluid is rubbed into the dressing until fully saturated (some products can be squeezed out). This equals approximately 2 mL for every 25 cm^2^ of wound area. Using less than the recommended quantity can reduce effectiveness and possibly cause adhesion to the wound. The fluid was applied to the wound area. The primary and secondary dressings were changed as needed. Daily changes were needed in highly exudating wounds. The dressings were changed every 2 or 3 days in the wounds, which did not exhibit signs of infection or excessive exudation. Antibiotics were used at the discretion of the attending physicians.

### 2.6. The Evaluated Characteristics

The wounds were evaluated during the dressing changes. Wound width and height were measured. Complete wound healing was characterised as 100% closure. The wound appearance was graded as mainly necrotic, sloughy, granulating, or epithelialising. Exudation was graded semiquantitatively as “no,” “low,” “medium,” or “high.” The presence of classically visible signs of infection was recorded. Subjective pain of the wound was evaluated with possible levels: no, low, medium, and high. 

### 2.7. Data Analysis

Wounds treated for at least two weeks were analysed for wound closure rate and size. Figure 1, Figure 2, Figure 3 and Figure 4 contain information from all patients. Patients who finished the study (completely healed or retained until week 12) were analysed for changes in wound bed appearance, exudation, the presence of classical visible signs of infection, and pain (Figure 1, Figure 2, Figure 3 and Figure 4), with respect to baseline with the Fisher exact test. Changes in wound sizes between particular weeks and baseline were evaluated with paired Wilcoxon rank-sum tests. 

## 3. Results

Over two-thirds of the DFU or VLU patients finished this study, while approximately half of the patients with PU or pSW were not evaluated at the end (Table 2). The high dropout rate in the PU group was caused by several reasons, none of which were prevalent. However, four of the patients with pSW were lost to follow-up after being discharged home. None of the deaths were related to the treatment. Treatment was changed for five of the patients at the discretion of the physicians since there was no improvement in healing or the patients felt irritation from the wounds and withdrew from the study. 

Of the patients who finished the study, the majority of the wounds had healed within 12 weeks (Table 3). Almost three-quarters of DFUs healed, followed by two-thirds of pSW, over half of VLU, and less than 50% of PU. 

The hard-to-heal wounds treated with HA-I gradually healed and became smaller, as indicated in Table 4 and Table 5. DFU healed fastest, followed by pSW, PU, and VLU. The rate of healing, expressed as a percentage of wound size reduction per day, increased during the 12 weeks of treatment, reaching about 6% size reduction per day in DFU and pSW. The healing rate in VLU and PU was positive, although smaller than that of DFU and pSW (Table 4). The medians of the sizes of the wounds decreased most significantly for DFU, followed by PU. Although the median size of PU did not decrease in weeks 10 and 12 compared to week 0, these ulcers were significantly smaller than their respective baseline values (Table 5).

Wound bed appearance significantly improved, mainly for DFU (*p* = 0.004), VLU (*p* = 0.002), and pSW (*p* = 0.013) (Figure 1). VLU and pSW were not sloughy at week 12. The wound bed improved in PU within 4–6 weeks and then stagnated. Some of the PU exhibited tissue necrosis even at later stages of treatment, and the change in wound bed appearance was not significant compared to the baseline (*p* = 0.25).

In terms of wound exudation, significant improvements with respect to baseline levels were observed in VLU (*p* = 0.007) and DFU (0.0002) (Figure 2). Exudation did not differ from the baseline in PU (*p* = 1). Concomitantly, wound exudation was the lowest in PU at the beginning of the study. Test statistics (Fisher exact test) were not significant (*p* = 0.19) in pSW. However, the majority of the pSW were without secretion at the end of the study, and in that respect, the pSW improved the most.

Regarding the classical visible signs of infection, more than half of DFU were classified as exhibiting infection (Figure 3), whereas a minority of other wound types were labelled as infected. Despite DFU exhibiting the highest proportion of infected wounds at baseline, DFU improved in this parameter most noticeably. After week 8, DFU and wounds did not exhibit the classical visible signs of infection. The use of antibiotics in the patients was at the discretion of the physicians. Therefore, the antimicrobial effect cannot be attributed solely to HA-I.

The most painful wounds before the treatment were DFU (Figure 4). In all wound types, there was gradual improvement, and the wounds were reported mainly as painless or low pain at week 12 (*p* = 0.005 in DFU; *p* = 0003 in VLU; *p* = 0.267 in PU; *p* = 0.001 in pSW). 

**Figure 1 pharmaceutics-15-02268-f001:**
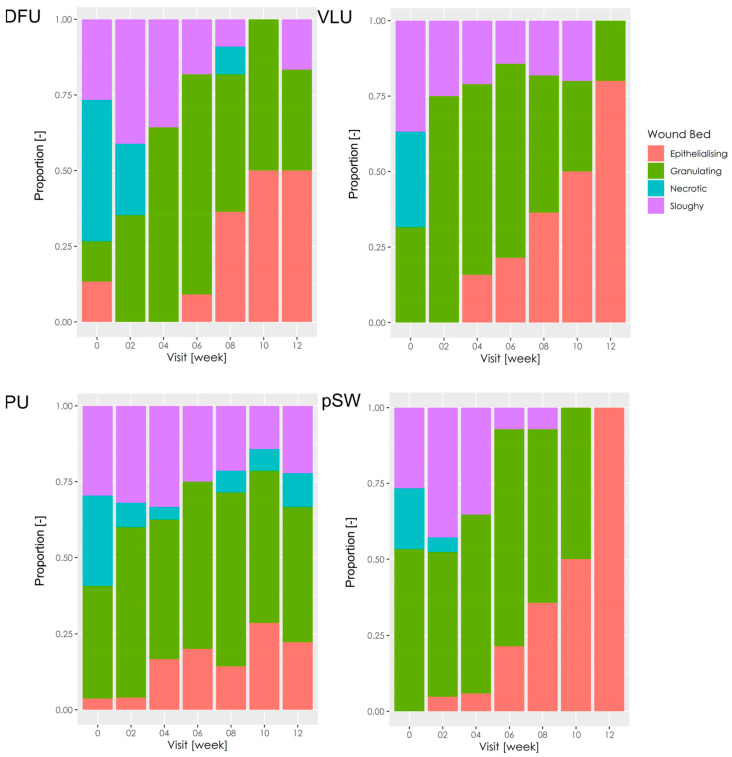
Wound bed appearance. The stacked bar charts show proportions of wounds showing epithelialisation, granulation, necrosis, or slough throughout this study period. DFU—diabetic foot ulcers; VLU—venous leg ulcers; PU—pressure ulcers; pSW—post-surgical wounds.

**Figure 2 pharmaceutics-15-02268-f002:**
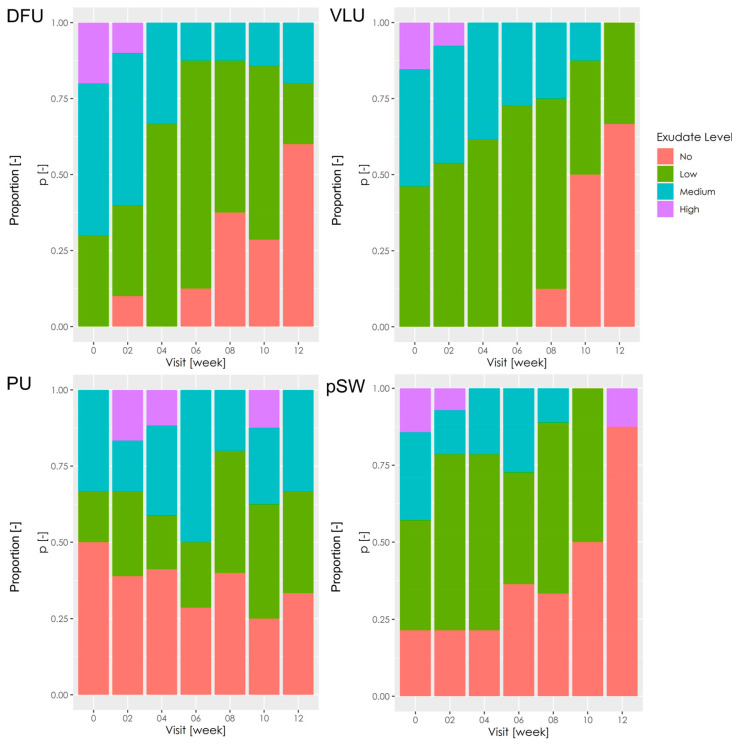
Wound exudation. The stacked bar charts show proportions of wounds showing exudation on a semiquantitative scale as “no,” “low,” “medium,” or “high.” DFU—diabetic foot ulcers; VLU—venous leg ulcers; PU—pressure ulcers; pSW—post-surgical wounds.

**Figure 3 pharmaceutics-15-02268-f003:**
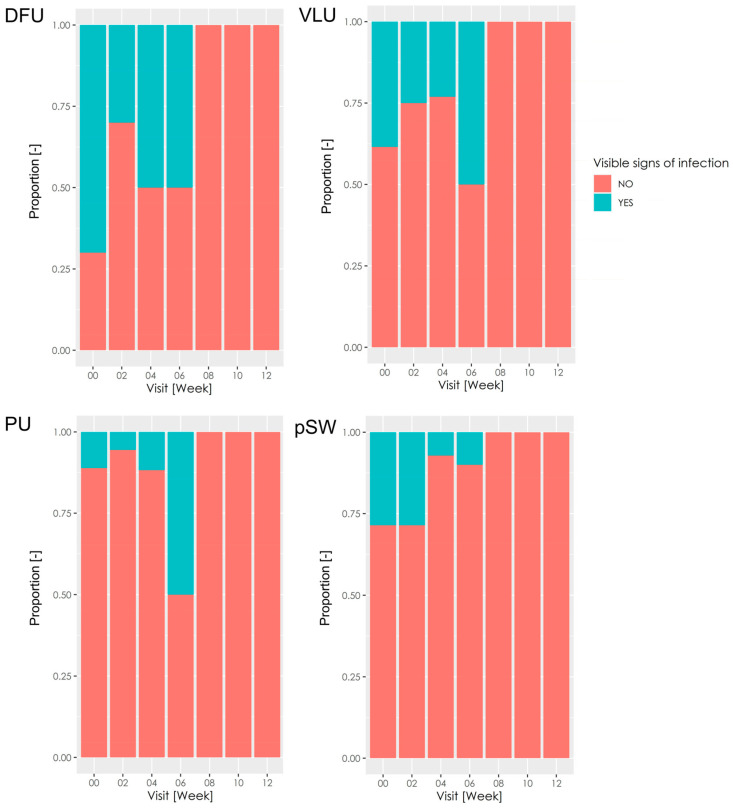
The classical visible signs of infection. The stacked bar charts show the proportion of wounds showing classically visible signs of infection. DFU—diabetic foot ulcer; VLU—venous leg ulcer; PU—pressure ulcer; pSW—post-surgical wound.

**Figure 4 pharmaceutics-15-02268-f004:**
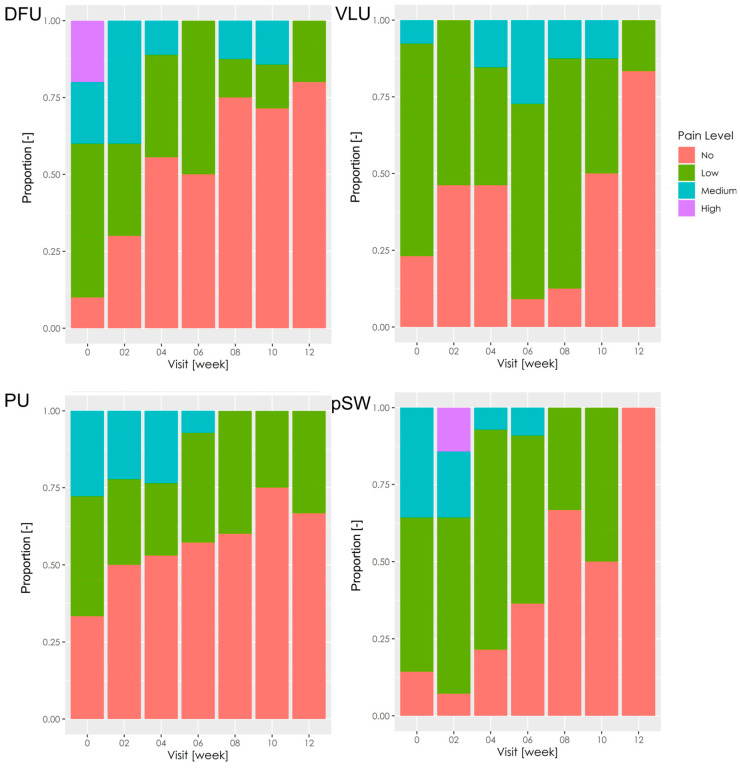
Pain between dressing changes. The stacked bar charts show the proportion of pain reported by the patients. DFU—diabetic foot ulcer; VLU—venous leg ulcer; PU—pressure ulcer; pSW—post-surgical wound.

## 4. Discussion

Wounds treated with hyaluronic acid-iodine complex fluid showed marked improvement in several parameters, leading to the complete healing of most hard-to-heal wounds. 

The wound healing rate was positive in all wound types and, therefore, shows that most wounds did not stagnate (Table 4). A combination of factors likely caused the increase in the wound healing rate during this study period. First, fewer keratinocytes are required to cover the same surface area in smaller wounds. Second, as a wound heals, infection recedes, exudation decreases, and there is more favourable (i.e., granulation) tissue to migrate over. The mean wound closure rate in our 12-week study was around 1.27% wound size reduction/day (Table 5). 

Since we investigated the applicability and efficacy of the hyaluronic acid + KI_3_ complex in hard-to-heal wounds with various aetiologies, our study did not include a placebo or standard of care. Diabetic wounds healed in 71.4% of cases (the baseline mean size of the DFU retained in this study was 35.8 ± 30.6 cm^2^ (Table 3)). Previous studies reported a 31.4% healing rate of DFU with saline (baseline ulcer size 1–25 cm^2^) [23], 41.7% with a standard of care (4.7 ± 2.9 cm^2^ at baseline) [24], or 73.7% with saline gauze (initial mean size 2.3 ± 2.7 cm^2^) [25]. It is evident that the healing time of DFU depends on its size, and regarding that, HA+ KI_3_ performed well. The duration of a hard-to-heal wound is one of the main factors driving the healing of chronic wounds [26]. Although the DFU in our study were regarded as chronic, we reached a similar wound healing rate as reported by Lafontaine et al., who showed complete healing in approximately 72% of silver- or placebo-treated acute DFU (present < 6 weeks) [27]. Sobotka et al. used HA-I to treat DFU, which healed completely in 6/15 (40%) patients within 12 weeks [19]. Another nine patients healed later. The onset of granulation was, on average, after 44 days of treatment. Clinical improvement was recorded for most of the wounds. 

Several studies observed a complete healing rate of VLU of 23–35% after 12 weeks of standard treatment [28,29,30,31,32]. We reached 55.6% completely healed wounds, which indicates support for wound healing compared to standard therapies (Table 3). A large RCT comparing cadexomer iodine and silver dressing in venous and mixed leg ulcers showed higher healing rates—64% and 63%, respectively [33]. The mean ulcer surface in that study was 7.04 ± 8.8 cm^2^, while VLU in our study were 26.3 ± 24.4 cm^2^ (plus one circumferential ulcer with no reported size). Since larger wounds heal longer, fewer VLUs in our study were expected to heal longer. Holloway et al. reported an epithelialisation rate of 0.95 ± 0.17 cm^2^/week in VLU treated with cadexomer iodine ointment [34], which is similar to HA-I in VLU (1.04 ± 0.77% wound size reduction/day, Table 4). 

Pressure ulcers healed the least of the hard-to-heal wounds included in our study (44.4% in 12 weeks, Table 3). However, when compared to healing rates evinced by standards of care—27.7% [35], 29% [36], the PU healed faster than expected. 

Post-surgical wounds healed in 62.5% of cases (Table 3). Three discharged patients with pSW who were lost to follow-up were healing well, which indicates that the healing rate could have been even higher. Sobotka et al. treated patients with abdominal dehiscences with HA-I [20]. These wounds were complicated by multiple fistulae. HA-I fluid aided clearance of the wounds before reconstructive surgery (3 patients), spontaneous healing (3 patients), or healing with skin grafting (1 patient). The mean time to surgical treatment of fistulae was 13.1 ± 3.8 weeks and 17.7 ± 10.5 weeks until complete healing after the surgeries. These numbers are not directly comparable with those in our study (Table 3). However, pSW in this study were less severe (anal fistula, post-amputation complications, and laparostomy).

Treatment with HA-I was generally well tolerated. Only five patients manifested increased sensitivity, swelling, or exudation, and treatment with HA-I was discontinued. Of those, one patient was highly sensitive to the following treatments with other non-iodine dressings; another patient was concomitantly treated with chemotherapeutics, which could have caused adverse skin changes. In most patients who responded well to the HA-I treatment, physicians reported fast necrosis resolution and granulation onset. Another advantage is its application form. The fluid is conveniently applied below undermined wound edges or into fistulae. The fluid maintains a moist environment, which perhaps supports the dissolution of necrotic tissue via autologous debridement. Hyaluronic acid supports migration in various cells, such as keratinocytes and fibroblasts [37,38]. Iodine limits bacteria and may reduce excessive protease activity in wounds [39]. 

This study is limited by relatively low numbers of patients dispersed throughout various aetiologies of hard-to-heal wounds. This makes the interpretation of the results complex since each wound type has its own specificity. We did not recruit just one type of wound because we were interested in the wound-healing effects of HA-I across different wound aetiologies. We also included non-healing post-surgical wounds because there were previous favourable outcomes in their treatment with HA-I. To aid in the interpretation of the data, we stratified the results by wound type. We did not add a standard of care to our comparisons. Therefore, we did not control for internal validity or external bias. However, we believe this study brings insight into the capabilities of HA-I, which will lead to the preparation of a controlled trial. 

## 5. Conclusions

Hyaluronan and iodine fluid aided the healing of 57.6% of hard-to-heal wounds of various aetiologies in 12 weeks and were well tolerated. It maintains a moist wound environment and is anti-adhesive. Its physical form as a viscous liquid enables it to treat irregularly shaped wounds and fistulae.

## Figures and Tables

**Table 1 pharmaceutics-15-02268-t001:** Patient and wound type characteristics.

Characteristic	Value	SD/%
Participant characteristics (n)	56	
Age, years, (mean ± SD)	67.86	14.46
Sex, female, n (%)	21	37.5%
Sex, male, n (%)	35	62.5%
Smoker, n (%)		
Yes	10	17.24%
Comorbidities, n (%)		
Diabetes mellitus type I	5	8.62%
Diabetes mellitus type II	22	37.93%
Hypertension	34	58.62%
Obesity	18	31.03%
Heart disease	23	39.66%
Ischaemic heart disease	16	27.59%
Angiopathy	6	10.34%
Other	22	37.93%
Wound type, n (%)		
DFU	10	17.24%
VLU	13	22.41%
PU	18	31.03%
pSW	15	24.14%
Infection, n (%)		
Yes	19	32.76%

DFU—diabetic foot ulcer; VLU—venous leg ulcer; PU—pressure ulcer; pSW—post-surgical wound.

**Table 2 pharmaceutics-15-02268-t002:** Patients who did not finish the study.

	DFU	VLU	PU	pSW
Discontinued treatment	1	1	2	1
Dismissed and lost to follow-up	1	1	2	4
Died	1	1	2	-
Unspecified	-	1	3	2
Total	30% (3/10)	30.8% (4/13)	50% (9/18)	46.7% (7/15)

DFU—diabetic foot ulcers, VLU—venous leg ulcers, PU—pressure ulcers, pSW—post-surgical wounds.

**Table 3 pharmaceutics-15-02268-t003:** Proportions of the wounds healed within 12 weeks.

	All Wounds	DFU	VLU	PU	pSW
Healed	57.6% (19/33)	71.4% (5/7)	55.6% (5/9)	44.4% (4/9)	62.5% (5/8)

DFU—diabetic foot ulcers, VLU—venous leg ulcers, PU—pressure ulcers, pSW—post-surgical wounds.

**Table 4 pharmaceutics-15-02268-t004:** Healing rate [% wound size reduction/day]. The overall rate was computed from the baseline and week 12 (or the latest known) sizes, whereas the values for particular weeks reflect the size difference from preceding weeks.

	All Wounds			DFU			VLU			PU			pSW		
	Mean	SD	n	Mean	SD	n	Mean	SD	n	Mean	SD	n	Mean	SD	n
Overall	1.27	1.06	56	1.62	1.19	10	1.04	0.77	13	1.22	1.11	18	1.22	1.23	15
Week 2	0.54	3.08	56	1.11	2.03	10	−0.44	2.93	13	1.39	2.41	18	−0.06	4.54	15
Week 4	2.15	2.38	52	2.54	1.43	9	1.99	2.11	11	1.58	2.50	17	2.28	2.73	15
Week 6	2.43	2.37	42	3.61	0.67	8	2.53	1.97	9	2.14	2.82	14	1.75	1.28	11
Week 8	2.63	2.78	33	3.38	2.85	8	3.13	3.13	6	2.42	3.11	10	2.06	2.66	9
Week 10	2.89	3.43	25	4.76	2.34	7	2.16	2.48	6	0.79	4.63	8	4.04	2.43	4
Week 12	4.38	3.19	17	5.69	1.71	5	4.00	3.40	4	2.53	4.09	6	6.51	0.90	2

DFU—diabetic foot ulcers, VLU—venous leg ulcers, PU—pressure ulcers, pSW—post-surgical wounds.

**Table 5 pharmaceutics-15-02268-t005:** Wound size during the treatment. The table shows medians (M) of approximated areas (length × width, cm^2^) for particular wound types, interquartile ranges (IQR), number of subjects (n), and statistical difference to baseline as * *p* < 0.05, ** *p* < 0.01, *** *p* < 0.001 (Wilcoxon paired rank sum test). When n < 3, the test was not computed.

	All Wounds				DFU				VLU				PU				pSW			
	M	IQR	*p*	n	M	IQR	*p*	n	M	IQR	*p*	n	M	IQR	*p*	n	M	IQR	*p*	n
Week 0	16.8	34	NA	56	30	30	NA	10	38	37	NA	13	11.3	18	NA	18	11.5	26	NA	15
Week 2	15	32	**	56	22	23	0.1	10	38	27	0.22	13	8.8	19	0.06	18	12.9	20	0.11	15
Week 4	8	24	***	52	11	14	**	9	32	45	0.11	11	5.0	10	**	17	4.8	22	0.05	15
Week 6	6	16	***	42	6.8	6.3	**	8	15	44	0.06	9	5.5	11	**	14	7.0	24	**	11
Week 8	5.3	18	***	33	4.5	5.1	**	8	4.5	55	0.05	6	7.1	9.8	**	10	14.4	23	*	9
Week 10	2	16	***	25	0.5	2.7	**	7	22	46	0.08	6	11.3	16	*	8	6.4	11	0.13	4
Week 12	1.1	21	***	17	0.5	0.8	*	5	17	36	0.06	4	12.1	19	*	6	1.0	1	NA	2

DFU—diabetic foot ulcers, VLU—venous leg ulcers, PU—pressure ulcers, pSW—post-surgical wounds. NA—not applicable.

## Data Availability

Data are unavailable due to privacy and ethical restrictions.

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
