# Peer review of "Clinical Efficacy of Hyaluronic Acid with Iodine in Hard-to-Heal Wounds"

_pharmaceutics, 2023, doi:10.3390/pharmaceutics15092268_

Round 1
Reviewer 1 Report
Interesting clinical study.
Well organized and presented, but the English needs moderate revising.
This is a very interesting clinical study. However, neither the title nor the abstract highlights this important feature. The type of this clinical study should also be stated in these two places to attract the readers.
Line 103: “The research was conducted as a multi-center, post-market clinical follow-up study (PMCF)” better reads: : A post-market clinical follow-up study (PMCF) in multi center was conducted”
Section : The evaluated characteristics”
Table 1, Organize the content and align left the first column
Throughout the manuscript, abbreviations in the tables legend look better if given as footnote
Figure 3. “The visible signs of infection” in the test this term is referred as “the classical visible signs”
Please be consistent to make it easier to the reader to follow the ideas
Figure 4. “Pain between dressing changes.” Would it be better to use the following “ Pain during dressing changes”??
English needs moderate revising to make sure he academic style is preserved
Author Response
Thank you for the time devoted to the review, and resulting suggestions and comments.
Well organized and presented, but the English needs moderate revising.
The English was improved.
This is a very interesting clinical study. However, neither the title nor the abstract highlights this important feature. The type of this clinical study should also be stated in these two places to attract the readers.
We added “clinical” both to the title and abstract.
Line 103: “The research was conducted as a multi-center, post-market clinical follow-up study (PMCF)” better reads: : A post-market clinical follow-up study (PMCF) in multi center was conducted”
The sentence was substituted with: “A post-market clinical follow-up study (PMCF) was conducted in multiple wound-care centers.”
Section : The evaluated characteristics”
Table 1, Organize the content and align left the first column
The journal uses tables with all fields centered. Therefore, we use it throughout the manuscript.
Throughout the manuscript, abbreviations in the tables legend look better if given as footnote
We agree, the abbreviations were moved to footnote sections.
Figure 3. “The visible signs of infection” in the test this term is referred as “the classical visible signs”
Please be consistent to make it easier to the reader to follow the ideas
We unified the term to “classical visible signs of infection“.
Figure 4. “Pain between dressing changes.” Would it be better to use the following “ Pain during dressing changes”??
Pain was recorded for the time between dressing changes, we did not evaluate pain during dressing changes.
Reviewer 2 Report
Comments to the authors:
1. How fluid was applied to the wound area is not obviously disclosed.
2. the function/mechanism of fluid exposure should be explained.
3. It is better to present the wound healing promotion by images.
4. The conclusion doesn't meet the main text. It should be reconsidered and written again.
Moderate editing of English language required.
Author Response
Thank you for your comments a recommendations.
- How fluid was applied to the wound area is not obviously disclosed.
We described the application more explicitly in the Treatment regiment section.
- the function/mechanism of fluid exposure should be explained.
We now discuss possible mechanisms of how the fluid supports wound healing.
- It is better to present the wound healing promotion by images.
We agree that the amount of information presented in the tables may seem overwhelming and some important details are lost compared to seeing images. However, we decided to give as much relevant information as possible in tables because the wound healing is represented most objectively given the various wound sizes, types and the number of patients. Conclusions and comparisons are also more readily done when comparing for example wound healing rates.
- The conclusion doesn't meet the main text. It should be reconsidered and written again.
We added some information into the discussion. Now all the information in the conclusion can be found throughout the manuscript.
Reviewer 3 Report
The suggestions/comments/annotations are inserted into the attached file.

Minor editing of English language required
Author Response
Thank you for your detailed work and the suggestions.
consider "led"
Changed.
consider "wounds healing."
Changed.
to be removed
Changed.
consider "functionality of this type of soft tissue."
Changed.
consider "on how"
Changed.
this assertion should be accompanied by a proper citation
Added.
nowhere in this reference is this property of iodine mentioned. this issue should be fixed/clarified.
We added another reference mentioning this property. The previous reference was left as it shows details of resistance mechanisms relevant to the other antiseptics mentioned in the sentence.
needs to be supported by corresponding citation\(s\)
The references were added.
needs to be supported by corresponding citation\(s\)
This was not measured and published. However regarding the iodine release, it may be slowed by the viscous nature of Hyiodine fluid. Regarding the oxidative potential of Hyiodine fluid - The oxidative potential of I2 in Hyiodine is „diluted“ by an effective reaction with the excess of I- giving I3- , where a formal oxidation state of the iodine atom is -1/3. While the oxidative potential of I2 in PVP is „diluted“ by less effective interaction of I2 with a lone electron pair of oxygen, where a formal oxidation state of the iodine atom remains 0. An atom with the formal oxidation state 0 is usually a stronger oxidant than an atom with the formal oxidation state -1/3.
We changed the statements to reflect that.
consider "triiodide"
Changed.
this reference has nothing to do with the associated text. needs to be clarified
We believe that the reference is appropriate since the review by Boateng and Catanzano is devoted to advanced wound dressings and mentions Hyiodine on page 3657 down left.
consider "PMA \(phorbol 12-myristate 13-acetate\)-"
Added.
to be specified the \(mean\) value
Added.
consider "area."
Added.
consider "every"
Changed
consider "exhibit"
Changed.
consider "gradually healed and became"
Changed.
consider "rate of healing, expressed as percentage of wound size reduction per day, increased"
Changed.
consider "pSWs \(Table 4\)."
Added.
consider "values \(Table 5\)."
Added.
consider "PU \(decubitus\)"
Changed
to avoid any confusion with the p qua\ ntity associated with the statistical tests considered, the authors are asked to change the symbol p on OY axes in Figures 1-4 with another one.
Changed
consider "acid-iodine"
Changed.
consider "surface area between"
Changed.
most likely this could be 1.27 cm2/day or cm2/week \(???\) because the values expresses a rate of w\ ound closure. secondly, is this mean value calculated from data listed somewhere in the paper. if yes, please, specify the place in the manuscript!
The unit was wrong and was changed and referenced to the corresponding table.
consider "KI3" instead of "KI3" throughout the manuscript!
Changed.
consider "cm2\) \(Table 3\)."
Added.
consider "therapies \(Table 3\)."
Added.
consider "VLU observed in this study \(1.04+-0.77 cm2/day or cm2/week ???\)."
Changed.
consider "weeks, Table 3\)."
Added.
consider "62.5% \(Table 3\)."
Added.
the data reported in ref. 17 seem not to match well at all with the value of 19.4+-16.3 weeks invoked by the authors of this study. needs to be clarified
We corrected the numbers and clarified the comparisons.
consider "viscous liquid"
Changed.
Reviewer 4 Report
This article is interesting as it discusses the application of the hyaluronic acid and iodine complex to treat hard-to-heal wounds of various etiologies. However, I have the following questions/ concerns:
- How did the authors determine doses of HA-I?
- Authors justified: “Since the study focused on treatment efficacy and applicability in various types of complicated wounds, there were no control treatments.” However, how do the authors justify HA-I fluid's toxic or adverse effect on adjacent normal cells without using controls?
- Authors are advised to explain the statement “deep and complicated wound defects” in the patients’ inclusion and exclusion criteria.
- Table 1 needs to be clarified. Can the authors explain it in a better manner?
- What is the rationale for selecting only female patients?
- Authors are advised to perform a correlation regression analysis among wound size, infection, pain intensity, and dose of HA-I.
- HA-I fluid was applied both directly and indirectly. Was there any difference in the effect of wound healing due to different ways of HA-I application?
- What is the molecular weight of HA used in preparing HA-I fluid?
- Authors are suggested to check the expression of a biomarker to evaluate wound healing.
Moderate editing of the English language may be required.
Author Response
Thank you for your comments and suggestions.
- How did the authors determine doses of HA-I?
We used 2 ml per 25 cm2 as recommended by the manufacturer.
- Authors justified: “Since the study focused on treatment efficacy and applicability in various types of complicated wounds, there were no control treatments.” However, how do the authors justify HA-I fluid's toxic or adverse effect on adjacent normal cells without using controls?
The HA-I fluid has CE certification which requires preclinical and clinical evaluation of its possible adverse effects. HA is regarded as safe and has no toxicity. Iodine has nonspecific broad-spectrum antimicrobial and cytotoxic effect depending on its dose. The dose does not elicit negative effects in human chronic wounds.
- Authors are advised to explain the statement “deep and complicated wound defects” in the patients’ inclusion and exclusion criteria.
The statement was specified.
- Table 1 needs to be clarified. Can the authors explain it in a better manner?
&
- What is the rationale for selecting only female patients?
We believe that Table 1 reflects a standard way how to present demographics, since it is used in several clinical studies. We included 35 male, and 21 female patients. We wanted to present the data as concisely as possible. Therefore, only one value in two-level variables (Sex, Smoking, Infection) is presented, since both numbers (male/female, yes/no) add to the total number 56. However, we added for clarity a row with the number of males included in the study.
- Authors are advised to perform a correlation regression analysis among wound size, infection, pain intensity, and dose of HA-I.
Such an analysis would bring insight in the scenario, where the dose of HA-I would change and one could investigate the dependence of the mentioned parameters on the HA-I fluid dose. However, the dose was proportional to wound sizes in this study. We will consider to investigate the influence of different doses of the fluid in the next studies.
- HA-I fluid was applied both directly and indirectly. Was there any difference in the effect of wound healing due to different ways of HA-I application?
The mean of application was at the discretion of the physicians based on the characteristics of the wound. The direct application was mainly to smaller or irregularly-shaped (deep) wounds. Indirect application was mainly in larger wounds, where gauze was conveniently soaked and ensured even spreading of the fluid over the wound. Therefore, there the two means of application can not be compared.
- What is the molecular weight of HA used in preparing HA-I fluid?
We added the value (> 1 MDa) into Materials section of the Materials and methods.
- Authors are suggested to check the expression of a biomarker to evaluate wound healing.
Thank you for the suggestion, we will consider sampling wound fluid and/or biopsies. However, such samples were not collected in this study.
Round 2
Reviewer 4 Report
The manuscript can be accepted.